

# A split-and-transfer flow based entropic centrality

Frédérique Oggier[1,*], Silivanxay Phetsouvanh[2] and Anwitaman Datta[2,*]

[1] Division of Mathematical Sciences, Nanyang Technological University, Singapore, Singapore
[2] School of Computer Science and Engineering, Nanyang Technological University, Singapore, Singapore
[*] These authors contributed equally to this work.

## ABSTRACT

The notion of entropic centrality measures how central a node is in terms of how uncertain the destination of a flow starting at this node is: the more uncertain the destination, the more well connected and thus central the node is deemed. This implicitly assumes that the flow is indivisible, and at every node, the flow is transferred from one edge to another. The contribution of this paper is to propose a split-and-transfer flow model for entropic centrality, where at every node, the flow can actually be arbitrarily split across choices of neighbours. We show how to map this to an equivalent transfer entropic centrality set-up for the ease of computation, and carry out three case studies (an airport network, a cross-shareholding network and a Bitcoin transactions subnetwork) to illustrate the interpretation and insights linked to this new notion of centrality.

## INTRODUCTION

Centrality is a classical measure used in graph theory and network analysis to identify important vertices. The meaning of "important" depends on the nature of the problem analyzed, e.g., hubs in networks, spreaders of a disease, or influencers in social networks. Commonly used centrality measures include: the *degree centrality* which is the degree (or in-degree/out-degree) of the vertex depending on whether the graph is directed, possibly normalized to get the fraction of vertices a given vertex is connected to; the *closeness centrality* which is the reciprocal of the sum of the shortest path distances from a given vertex to all others, typically normalized, and indicates how close a given vertex is to all other vertices in the network; the *betweenness centrality* which is the sum of the fraction of all pairs of shortest paths that pass through it, indicating the extent to which a given vertex stands between other vertex pairs (see e.g., *Estrada, 2011* for a survey of different centrality measures and how centralities fit into the more general framework of complex networks). These were extended to weighted graphs, though at the risk of changing the interpretation of the measure, e.g., one may use weighted degrees instead of degrees, but this measure does not count the number of neighbors anymore (see e.g., *Opsahl, Agneessens & Skvoretz, 2010* for a discussion on using the above cited centrality measures for weighted graphs). Another way to determine centrality is to assign as centrality a (scaled) average

Corresponding author
Anwitaman Datta,
anwitaman@ntu.edu.sg

of the centralities of the neighbors. This is the idea behind *eigenvector centrality* discussed by *Newman (2009)*, which was already debated by *Bonacich (1972)*, who later generalized it to *alpha centrality* (*Bonacich & Lloyd, 2001*). Alpha centrality introduces an additive exogenous term, which accounts for an influencing factor which does not depend on the network structure. Though *Katz centrality* (*Katz (1953)*) relies on the idea that importance is measured by weighted numbers of walks from the vertex in question to other vertices (where longer walks have less weights than short ones), it turns out that the alpha centrality and Katz centrality differ by a constant term. With these three centralities, a highly central vertex with many links tends to endorse all its neighbors which in turn become highly central. However one could argue that the inherited centrality should be diluted if the central vertex is too magnanimous in the sense that it has too many neighbors. This is solved by Page Rank centrality, which is based on the *PageRank* algorithm developed by *Page et al. (1999)*. *Iannelli & Mariani (2018)* proposed ViralRank as a new centrality measure, defined to be the average random walk effective distance to and from all the other nodes in the network. This measure is meant to identify influencers for global contagion processes. *Benzi & Klymko (2015)* showed that a parameterized random walk model can capture the behavior of a gamut of centrality measures, including degree centrality (walks of length one) and eigenvector based centrality models (considered as infinite walks), which contain the eigenvector and Katz centralities as particular cases. This parameterized model helps explain and interpret the high rank correlation observed among degree centrality and eigenvector based centralities. *Schoch, Valente & Brandes (2017)* argues that the role of the network structure itself should not be underestimated when looking at correlations among centralities.

Notwithstanding this high rank correlation among centrality measures, each measure captures the vertex importance subject to a certain interpretation of importance, which is a key rationale behind studying different centrality models in different contexts. A seminal work by *Borgatti (2005)* looked at which notion of centrality is best suited given a scenario, by characterizing the scenario as a flow circulating over a network: a typology of the flow process is given across two dimensions, the type of circulation (parallel/serial duplication, transfer) and the flow trajectories (geodesics, paths, trails, or walks): a flow may be based on *transfer*, where an item or unit flows in an indivisible manner (e.g., package delivery), or by serial replication, in which both the node that sends the item and the one that receives it have the item (e.g., one-to-one gossip), or parallel duplication, where an item can be transmitted in parallel through all outgoing edges (e.g., epidemic spread). It was shown for example that betweenness is best suited for geodesics and transfer, while eigenvector based centralities should be used for walks and parallel duplication. Indeed, betweenness is based on shortest paths, suggesting a target to be reached as fast as possible, and thus fitting transfer. Using Katz's intuition, eigenvector based centralities count possible unconstrained walks, and they are consistent with a scenario where every vertex influences all of its neighbors simultaneously, which is consistent with parallel deduplication. This flow characterization is of interest for this work, since we will be looking at a case where a flow is actually not just transferred, but also split among outgoing edges, with the possibility to partly remain at any node it encounters. This scenario could typically be motivated by

financial transactions, which are transferred, not duplicated. However when transferred, the flow of money is not indivisible. Based on Borgatti's typology, a measure of centrality for transfer should be based on paths rather than eigenvectors. This is indeed the approach that we will explore.

Our starting point is the notion of entropic centrality as proposed by *Tutzauer (2007)*. A (directed) graph $G = (V, E)$ with vertex set $V$ and edge set $E$ is built whose edges are unweighted. To define the centrality of $u \in V$, the probability $p_{u,v}$ that a random walk constrained to not revisit any vertex (thus, only forming paths) starting at $u$ terminates at $v$ is computed. To model the stoppage of flow/walk at any vertex, an edge to itself (self-loop) is added. The process of computing $p_{u,v}$ is thus to consider a constrained random walk to start at node $u$, and at every node $w$ encountered in the path, to choose an outgoing edge uniformly at random among the edges leading to unvisited nodes (or choosing the self-loop to terminate the walk). Then the entropic centrality $C_H(u)$ of $u$ is defined to be

$$C_H(u) = -\sum_{v \in V} p_{u,v} \log_2 p_{u,v}. \tag{1}$$

This notion of entropic centrality was adapted in *Nikolaev, Razib & Kucheriya (2015)* to fit a Markov model, where instead of paths, unconstrained random walks are considered, for computational efficiency. In general, how to compute centrality at scale is an interesting direction of study in its own right, e.g., *Fan, Xu & Zhao (2017)*, but this is somewhat orthogonal to the emphasis of the current work.

In this work we revisit and generalize the original concept of entropic centrality to make it more flexible. To do so, we first interpret the "transfer" centrality proposed in *Tutzauer (2007)* as having (1) an underlying graph, where every edge has a probability which is that of being chosen uniformly at random among the other outgoing edges of a given vertex, and (2) an indivisible flow which starts at a vertex $u$, and follows some path where the probability to choose an edge at every vertex in this path is given by the probability attached to this edge, taking into account unvisited neighbors, to reach $v$. Since the flow is indivisible, the self-loop represents the probability for this flow to stop at a given vertex.

In our generalization, we similarly assume that we have (1) an underlying graph, only now the probability attached to each edge depends on the scenario considered and could be arbitrary, (2) the flow used to measure centrality can split among neighbors, by specifying which subsets it goes to with which probability, at every vertex it encounters (as per a *flow* in the traditional network analysis sense, flow conservation applies, meaning that the amount of flow that goes out of $u$ is the same amount of flow that reaches all of its neighbors). Again, a self-loop is an artifact introduced to capture the effect of the flow on vertices, even if none of the flow actually remains in the vertex (As in *Nikolaev, Razib & Kucheriya, 2015*, a zero probability would otherwise render zero contribution to the entropic centrality calculation). While the underlying phenomenon may have self-loops, they may or not be directly used to determine the self-loops needed for the mathematical model. This should be determined based on the scenario being modeled.

The above motivates the notion of a *split-and-transfer* entropic centrality. Since propagation of flow is an indicator of spread over the network, we will also consider a scaled

version of entropic centrality, where a multiplicative factor is introduced to incorporate additional information, which may suggest an a priori difference of importance among the vertices, for instance, if the data suggests that some vertices handle volume of goods much larger than other vertices.

The contributions of this work are to (1) introduce the above framework for split-and-transfer entropic centrality, (2) show in 'The transfer entropic centrality' that transfer centrality can be easily extended to consider arbitrary probabilities on graph edges and (3) prove that computing the split-and-transfer entropic centrality can be reduced to transfer entropic centrality over a graph with suitable equivalent edge probabilities (which is crucial from a practicality perspective), as shown in Proposition 1 of 'The transfer entropic centrality'. Studies that showcase and explore our technique are provided in 'Case Studies': (i) a cross-shareholding network representing portfolio diversification, that illustrates the versatility of our framework (ii) a subgraph of wallet addresses from the Bitcoin network, which originally motivated the study of split-and-transfer flows, and (iii) an airport network. Comparisons with other standard centralities (alpha, Katz, betweenness and PageRank) are given, showing that the entropic centrality captures different features.

# THE NOTION OF SPLIT-AND-TRANSFER ENTROPIC CENTRALITY

## The transfer entropic centrality

Consider the network shown on Fig. 1A and assume that the probability of an indivisible flow going from one vertex to another is uniform at random (including the option to remain at the current vertex). For a flow starting at $v_1$, there is then a probability $\frac{1}{4}$ to go to $v_4$, and a probability $\frac{1}{2}$ to continue to $v_5$, so the probability to go from $v_1$ to $v_5$ following the path $(v_1, v_4, v_5)$ is $\frac{1}{8}$. But since it is also possible to reach $v_5$ from $v_1$ using $v_3$ instead, an event of probability $\frac{1}{8}$, we have that the probability $p_{v_1,v_5}$ for an indivisible flow to start at $v_1$ and stop at $v_5$ is $p_{v_1,v_5} = \frac{1}{4}$. Similarly, we compute $p_{v_1,v_1}, p_{v_1,v_2}, p_{v_1,v_3}$ and $p_{v_1,v_4}$, and the transfer entropic centrality $C_H(u)$ of $u = v_1$ is $C_H(v_1) = \frac{3}{4}\log_2 4 + \frac{2}{8}\log_2(8) = 2.25$ by (1).

For a point of comparison, on the right of the same figure, we change the probability to go out of $v_1$, such that the edge $(v_1, v_2)$ is chosen with a probability $\frac{1}{2}$, while the probability is $\frac{1}{6}$ for using the edges to the other vertices (including a probability $\frac{1}{6}$ that the flow just stays at $v_1$ itself). The resulting probabilities are provided on Fig. 1B. There is no complication in computing $C_H(v_1)$ using (1) with non-uniform probabilities. This reduces slightly the centrality of $v_1$, which is consistent with the interpretation of entropic centrality: the underlying notion of entropy is a measure of uncertainty (*Tutzauer, 2007*), the uncertainty of the final destination of a flow, knowing that it started at a given vertex. Imagine the most extreme case where the edge $(v_1, v_2)$ is chosen with a probability 1, then even though $v_1$ has three potential outgoing neighbors, two of them are used with probability 0, so the centrality of $v_1$ would reduce considerably, as expected, since there is no uncertainty left regarding the destination of a flow at $v_1$.

The notion of transfer entropic centrality captured by (1) assumes that there is no vertex repetition in the paths taken by the flow. Figure 2 illustrates this hypothesis. Again for

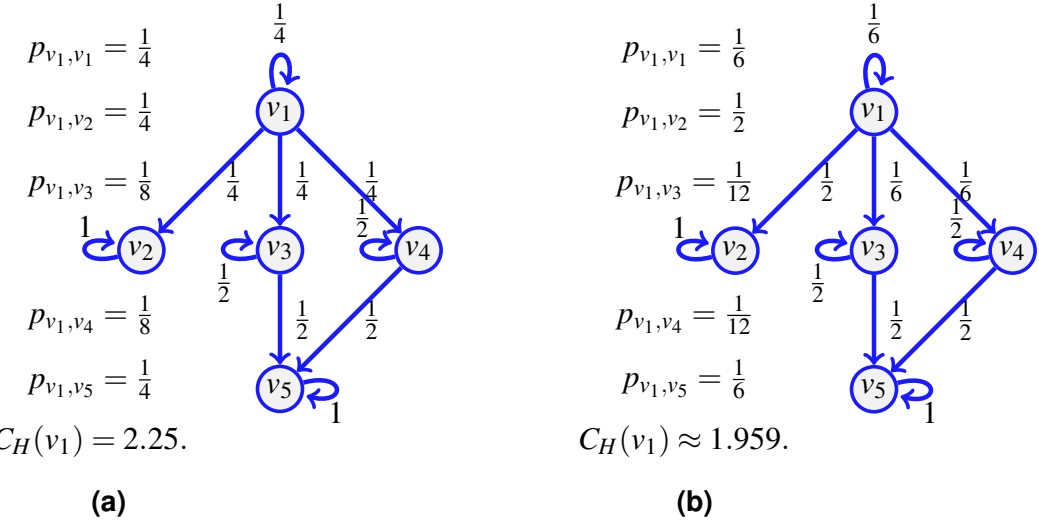

**Figure 1** The transfer entropic centrality $C_H(v_1)$ of $v_1$ is computed using (1), for a uniform edge distribution (the choice of an edge at a given vertex is chosen uniformly at random among choices of unvisited neighbors) in (A), and for a non-uniform distribution in (B).

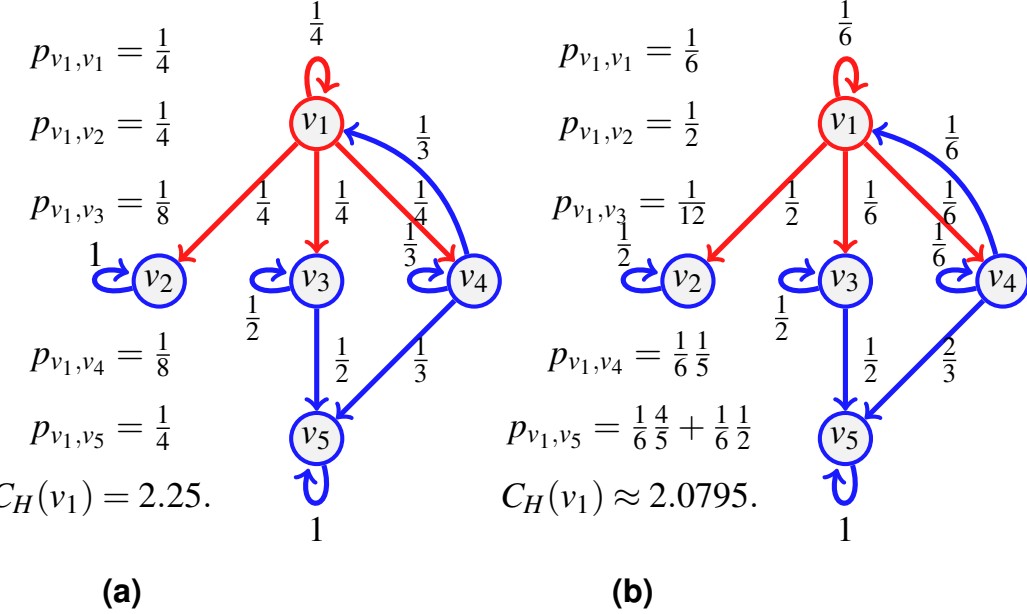

**Figure 2** An example of transfer centrality involving already visited neighbors. If probabilities are uniform at random (A), they are scaled according to the number of unvisited neighbors. If not (B), they are scaled proportionally to the existing probabilities.

the centrality of $v_1$, a flow leaves $v_1$, it can go to either $v_2$, $v_3$ or $v_4$. When reaching $v_4$, the flow cannot go back to $v_1$, since $v_1$ is already visited (and going back would not give a path anymore), there the probabilities to stay at $v_4$ and to go to $v_5$ from $v_4$ are modified.

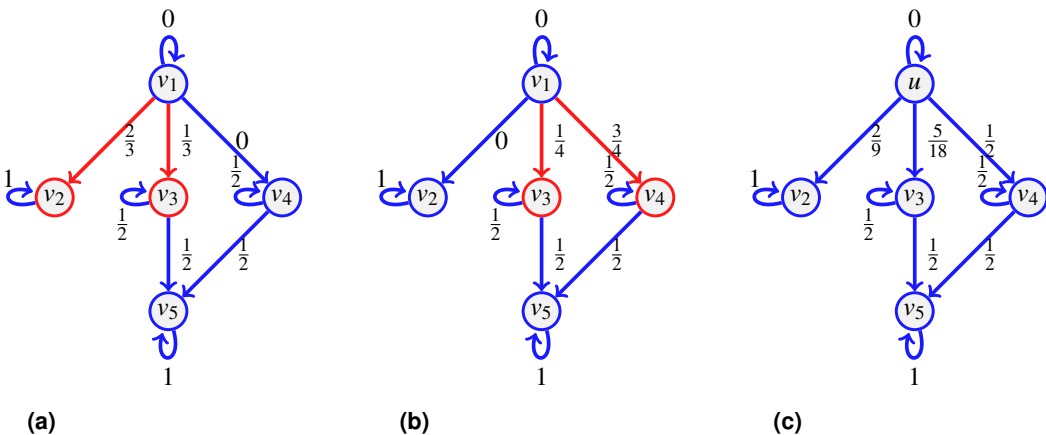

**Figure 3** An example of split-and-transfer entropic centrality: on (A) in red, the event corresponding to choosing $\{v_2, v_3\}$, in (B), the event $\{v_3, v_4\}$. The probabilities $p_{u,v}$ are computed by summing over both events, weighted by the respective event probability: $p_{v_1,v_2} = \frac{1}{3}(\frac{2}{3}) = \frac{2}{9}$, $p_{v_1,v_4} = \frac{2}{3}(\frac{3}{8}) = \frac{1}{4}$, $p_{v_1,v_3} = \frac{1}{3}(\frac{1}{6}) + \frac{2}{3}(\frac{1}{8})$, $p_{v_1,v_5} = \frac{1}{3}(\frac{1}{6}) + \frac{2}{3}(\frac{1}{8} + \frac{3}{8})$. This gives $C_H(v_1) \approx 1.9076$.

On the left, when probabilities are uniform, since now only two outgoing edges of $v_4$ are available, namely edges $(v_4, v_4)$ and $(v_4, v_5)$, each is assigned a probability of $\frac{1}{2}$. On the right, when probabilities are not uniform, we distribute the probability of going to some visited vertex proportionally to the rest of the available edges. Since $\frac{4}{6}$ is going to $v_5$ while $\frac{1}{6}$ is staying at $v_4$, we have 4 and 1 out of 5 respectively leaving and staying, thus obtaining the renormalized probabilities as $\frac{4}{6} + \frac{4}{5}\frac{1}{6} = \frac{4}{5}$ and $\frac{1}{6} + \frac{1}{5}\frac{1}{6} = \frac{1}{5}$.

The examples of Figs. 1 and 2 illustrate diverse cases of indivisible flow. By definition of indivisibility, the choice of an edge at a vertex $u$ corresponds to choosing subsets containing one vertex only in the list of all subsets of neighbors. We can thus set a probability 0 to all subsets which contain more than one vertex. Therefore, the definition of entropic centrality in (1), with or without uniform edge probabilities, are particular cases of the proposed split-and-transfer framework, that we discuss next.

## The split-and-transfer entropic centrality

Consider the network of Fig. 3 depicting a seller $v_1$ whose direct customers are $v_2, v_3, v_4$. Say we further know that when $v_1$ distributes a new batch of items, he does so to either customers $\{v_2, v_3\}$ or $\{v_3, v_4\}$, and in fact, the pair $\{v_3, v_4\}$ is preferred (they receive 2/3 of the batches, versus 1/3 for the group $\{v_2, v_3\}$). Furthermore, in the first case, $v_2$ receives a higher volume than $v_3$ (say 2/3 of the batch goes to $v_2$), while for the second case, $v_4$ takes 3/4 of the batch shared with $v_3$. Once $v_3, v_4$ obtain the items, they typically keep half for themselves, and distribute the other half to $v_5$.

To compute the centrality of $v_1$, we consider a divisible flow starting at $u = v_1$ which can split among different paths instead of following one. To model the choice of splitting among possible neighbors, we first define a probability $q(x)$ over the set $\mathcal{E}_u = \{ \{v_1\}, \{v_2\}, \{v_3\}, \{v_4\}, \{v_1, v_2\}, \{v_1, v_3\}, \{v_1, v_4\}, \{v_2, v_3\}, \{v_2, v_4\}, \{v_3, v_4\}, \{v_1, v_2, v_3\}, \{v_1, v_2, v_4\}, \{v_1, v_3, v_4\}, \{v_2, v_3, v_4\}, \{v_1, v_2, v_3, v_4\} \}$ such that, for our example, $q(\{v_2, v_3\}) = \frac{1}{3}$, $q(\{v_3, v_4\}) = \frac{2}{3}$, and

$q(x) = 0$ for other choices of $x$ (in contrast to *Oggier, Silivanxay & Datta (2018)* where it was chosen to be uniformly at random). This represents the fact that $1/3$ of the time, $v_1$ sends the goods to the pair $\{v_2, v_3\}$ (as shown in Fig. 3A), while for the rest of the time, it sends it to the pair $\{v_3, v_4\}$ (shown in Fig. 3B). We compute the path probabilities for each event, for $q(\{v_2, v_3\}) = \frac{1}{3}$ and for $q(\{v_3, v_4\}) = \frac{2}{3}$ accordingly.

We have further information: when $v_1$ deals with $\{v_2, v_3\}$, there is a bias of $\frac{2}{3}$ for $v_2$ compared to $\frac{1}{3}$ for $v_3$, and the bias is of $\frac{3}{4}$ for $v_4$ in the other case. The corresponding probabilities are attached to the edges $\{(v_1, v_2), (v_1, v_3)\}$ and $\{(v_1, v_3), (v_1, v_4)\}$ respectively (shown in Fig. 3C). Now that the edge probabilities are defined, we can compute the path probabilities. For example, from $v_1$ to $v_5$, we sum up the path probabilities for both events, weighted by the respective event probability: $\frac{1}{3}(\frac{1}{6}) + \frac{2}{3}(\frac{1}{8} + \frac{3}{8})$.

We next provide a general formula. We let a flow start at a vertex whose centrality we wish to compute, and at some point of the propagation process, a part $f_u$ of the flow reaches $u$. Let $\mathcal{N}_u$ be the neighborhood of interest given $f_u$, that is, the set of outgoing neighbors which have not yet been visited by the flow. Every outgoing edge $(u, v)$ of $u$ exactly corresponds to some outgoing neighbor $v$, so in what follows, we may refer to either one or the other. Let $\mathcal{E}_u$ denote the set of possible outgoing edge subsets (where every edge $(u, v)$ is represented by $v$ the neighbor). We attach a possibly distinct probability $q(x)$ to every choice $x$ in $\mathcal{E}_u$. Then $\sum_{x \in \mathcal{E}_u} q(x) = 1$.

Every $x$ in $\mathcal{E}_u$ corresponds to a set of edges $(u, v)$ for $v$ a neighbor. We further attach a weight $\omega_x(u, v)$ to every edge in $x$, with the constraint that $\sum_{(u,v) \in x} \omega_x(u, v) = f_u$. For example, we could choose all edges with equal weight, that is $\omega_x(u, v) = \frac{f_u}{i}$ for every $(u, v)$ in $x$ containing $i$ edges, to instantiate the special case where the flow is uniformly split among all edges.

For a given node $u$, we compute the expected flow from $u$ to a chosen neighbor $v$. Every such choice of $x$ comes with a probability $q(x)$, and every edge $(u, v)$ in $x$ has a weight $\omega_x(u, v)$, which sums up to

$$f_{uv} = \sum_{x \in \mathcal{E}_{u,v}} q(x)\omega_x(u, v), \tag{2}$$

where $\mathcal{E}_{u,v}$ contains the sets in $\mathcal{E}_u$ themselves containing $v$.

**Example 1** Consider the running example, with $u = v_1$. The set of neighbors of $u$ is $\mathcal{N}_u = \{u, v_2, v_3, v_4\}$. We assign the following probabilities: $q(\{u\}) = q_1$, $q(\{v_2\}) = q_2$, $q(\{v_3\}) = q_3$, $q(\{v_4\}) = q_4$, $q(\{u, v_2\}) = q_5$, $q(\{u, v_3\}) = q_6$, $q(\{u, v_4\}) = q_7$, $q(\{v_2, v_3\}) = q_8$, $q(\{v_2, v_4\}) = q_9$, $q(\{v_3, v_4\}) = q_{10}$, $q(\{u, v_2, v_3\}) = q_{11}$, $q(\{u, v_2, v_4\}) = q_{12}$, $q(\{u, v_3, v_4\}) = q_{13}$, $q(\{v_2, v_3, v_4\}) = q_{14}$, $q(\{u, v_2, v_3, v_4\}) = q_{15}$, with $\sum_{i=1}^{15} q_i = 1$. We write down explicitly the terms involved in the sum (2) for two nodes, $v_2$ and $v_3$:

$$f_{u,v_2} = q_2 f_u + q_5 \omega_{\{u,v_2\}}(u, v_2) + q_8 \omega_{\{v_2,v_3\}}(u, v_2) + q_9 \omega_{\{v_2,v_4\}}(u, v_2) + q_{11} \omega_{\{u,v_2,v_3\}}(u, v_2)$$
$$+ q_{12} \omega_{\{u,v_2,v_4\}}(u, v_2) + q_{14} \omega_{\{v_2,v_3,v_4\}}(u, v_2) + q_{15} \omega_{\{u,v_2,v_3,v_4\}}(u, v_2).$$
$$f_{u,v_3} = q_3 f_u + q_6 \omega_{\{u,v_3\}}(u, v_3) + q_8 \omega_{\{v_2,v_3\}}(u, v_3) + q_{10} \omega_{\{v_3,v_4\}}(u, v_3)$$
$$+ q_{11} \omega_{\{u,v_2,v_3\}}(u, v_3) + q_{13} \omega_{\{u,v_3,v_4\}}(u, v_3) + q_{14} \omega_{\{v_2,v_3,v_4\}}(u, v_3) + q_{15} \omega_{\{u,v_2,v_3,v_4\}}(u, v_3).$$

Then $f_{u,u} + f_{u,v_2} + f_{u,v_3} + f_{u,v_4} = f_u \sum_{i=1}^{15} q_i = f_u$. By setting $q_8 = \frac{1}{3}$ and $\omega_{\{v_2,v_3\}}(u,v_2) = f_u \frac{2}{3}$, we find $f_{u,v_2} = f_u \frac{2}{9}$. Also, adding up $q_{10} = \frac{2}{3}$ and $\omega_{\{v_2,v_3\}}(u,v_3) = f_u \frac{1}{3}$, $\omega_{\{v_3,v_4\}}(u,v_3) = f_u \frac{1}{4}$, we find $f_{u,v_3} = f_u \frac{1}{9} + f_u \frac{1}{6} = f_u \frac{5}{18}$. Similarly $f_{u,v_4} = f_u \frac{1}{2}$ and indeed $f_u \frac{2}{9} + f_u \frac{5}{18} + f_u \frac{1}{2} = f_u$.

We repeat the computations for $f_{v_3,v_5}$ and $f_{v_4,v_5}$. For that, we need to know what is $f_{v_3}$ and $f_{v_4}$, but in this case, since both $v_3$ and $v_4$ only have one incoming edge, we have that $f_{v_3} = f_{v_1,v_3}$ and $f_{v_4} = f_{v_1,v_4}$:

$$f_{v_3,v_5} = \frac{1}{2} f_{v_3} = f_u \frac{1}{2} \frac{5}{18}, f_{v_4,v_5} = \frac{1}{2} f_{v_4} = f_u \frac{1}{2} \frac{1}{2}, f_{v_5} = f_{v_3,v_5} + f_{v_4,v_5} = f_u \frac{7}{18}.$$

It is true that by setting $f_u = 1$, we have $f_{u,v_2} = \frac{2}{9} = p_{u,v_2}$ as computed in Fig. 3, but this is true because $p_{v_2,v_2} = 1$. If we consider $v_3$ instead, we find $f_{u,v_3} = \frac{5}{18} = 2p_{u,v_3}$, this is because we have computed what reaches $v_3$, but since $v_3$ has an outgoing edge, we need to distinguish what stays from what continues. Notice that by setting $f_u = 1$ and $f_{v_3} = f_{v_4} = 1$, we get

$$f_{u,v_2} = \frac{2}{9}, f_{u,v_3} = \frac{5}{18}, f_{u,v_4} = \frac{1}{2}, f_{v_3,v_5} = \frac{1}{2}, f_{v_4,v_5} = \frac{1}{2}.$$

We then assign to edge $(v_i,v_j)$ the probability $f_{v_i,v_j}$ (with $f_u = 1$) as reported on Fig. 3A.

The property of flow conservation observed in the example holds true in general, which we shall prove next. Indeed, when $v$ varies in $\mathcal{N}_u$, the sets $\mathcal{E}_{u,v}$ appearing in the summation $\sum_{v \in \mathcal{N}_u} \sum_{x \in \mathcal{E}_{u,v}} q(x)\omega_x(u,v)$ may intersect, so for each choice $x$, one can gather all the $\mathcal{E}_{u,v}$ that contains $x$. For this $x$, we find a term in the above sum of the form $q(x) \sum_{(u,v) \in x} \omega_x(u,v) = q(x)f_u$. Then

$$\sum_{v \in \mathcal{N}_u} \sum_{x \in \mathcal{E}_{u,v}} q(x)\omega_x(u,v) = \sum_{x \in \mathcal{E}_u} q(x)f_u = f_u.$$

This shows that the flow from $u$ to $v$ is conserved over all the neighbors $v \in \mathcal{N}_u$ given $f_u$. Thus, by setting $f_u = 1$, the quantity

$$f_{uv} = \sum_{x \in \mathcal{E}_{u,v}} q(x)\omega_x(u,v)$$

becomes a probability, and in fact, putting this probability on the edge $(u,v)$ in the context of the transfer entropic centrality gives the same result as the above computations using the split-and-transfer model, as in fact already illustrated on the figure in Example 1, since the probabilities displayed on the edges of the graph have been computed in this manner. We summarized what we computed in the proposition below.

**Proposition 1** *The split-and-transfer entropic centrality $C_{H,p}(u)$ of a vertex $u$ is given by*

$$C_{H,p}(u) = -\sum_{v \in V} q_{uv} \log_2(q_{uv})$$

*where $q_{uv} = \sum_{x \in \mathcal{E}_{u,v}} q(x)\omega_x(u,v)$ is computed from (2) with $f_u = 1$ and the usual convention that $0 \cdot \log_2 0 = 0$ is assumed. The index $p$ in $C_{H,p}$ emphasizes the dependency on the choice of the probability distribution $p$. Then we have $C_{H,unif}$ when $p$ is uniform as a particular case.*

We thus showed that the split-and-transfer entropic centrality is equivalent to a transfer entropic centrality, assuming the suitable computation of edge probabilities.

The notion of split-and-transfer entropic centrality characterizes the *spread* of a flow starting at $u$ through the graph. Now two vertices may have the same spread, but one vertex may be dealing with an amount of goods much larger than the other. In order to capture the *scale* of a flow, we also propose a scaled version of the above entropy.

**Definition 1** The scaled split-and-transfer entropic centrality is accordingly given by $F(f_u)C_{H,p}(u)$ where $F$ is a scaling function.

As a corollary, the computational complexity of this centrality measure is the same as that of *Tutzauer (2007)*, namely that of a depth first search, i.e., $O(|V|+|E|)$ (*Migliore, Martorana & Sciortino, 1990*). When the graph becomes large and some probability become negligible, a natural heuristic of setting them to 0 is used.

The scaling function $F$ may depend on the nature of the underlying real world phenomenon being modeled by the graph, with $F(f_u) = f_u$ being a simple default possible choice (other standard choices are $F(f_u) = \sqrt{f_u}$ or $F(f_u) = \log(f_u)$). We use the default choice in the example below.

**Example 2** Continuing with the same example, we use the edge probabilities as obtained in Example 1 to compute the transfer entropic centrality from Definition 1. The scaled entropic centralities of $u = v_1$ and $v_3$ are simply $f_u C_{H,p}(v_1) \approx f_u 1.9076$ and $f_{v_3} C_{H,p}(v_3) = f_{v_3}(\frac{1}{2}\log_2(2) + \frac{1}{2}\log_2(2)) = f_{v_3}$. Without the scaling factor, $C_{H,p}$ is a measure of spread, and it makes sense that $C_{H,p}(v_1) > C_{H,p}(v_3)$. However if $v_1$ is actually distributing some items in overall small amounts, while $v_3$ is not only getting this item from $v_1$ but also producing it and furthermore sending it only to $v_5$ but in large amounts, then the scaling factor could be used to refine the analysis and account for this extra information. From the moment $f_{v_3} \geq 1.9076 f_{v_1}$, $v_3$ will be deemed more important than $v_1$ as per the scaled entropic centrality measure.

## CASE STUDIES

### Shareholding in Tehran Stock Market

We next consider 131 companies from the Tehran Stock Market, as listed in Appendix A of *Dastkhan & Gharneh (2016)*.[1] They form the vertices of a cross-shareholding network of companies which have shares of other companies. There is an edge between $i$ and $j$ if company $i$ belongs to the investment portfolio of company $j$, i.e., $j$ owns some share of $i$. Therefore the in-degree of node $j$ is the number of companies that belong to the investment portfolio of company $j$. Conversely, the out-degree of node $j$ is the number of companies that are shareholders of $j$. Edges are weighted, edge $(i, j)$ has for weight the percentage of shares that company $j$ has from company $i$. We will consider this graph, shown in Fig. 4, and the graph with reverse edge directionality.

Nodes highlighted in green in Fig. 4 have one edge with weight more than 0.5, meaning that more than 50% of their shares are owned by another company, otherwise they are in grey. Nodes highlighted in vermillion, superseding the other coloring, have the highest in-degrees, which means that they own shares of many other companies. They are nodes

[1]We thank the authors of *Dastkhan & Gharneh (2016)* for sharing their data with us.

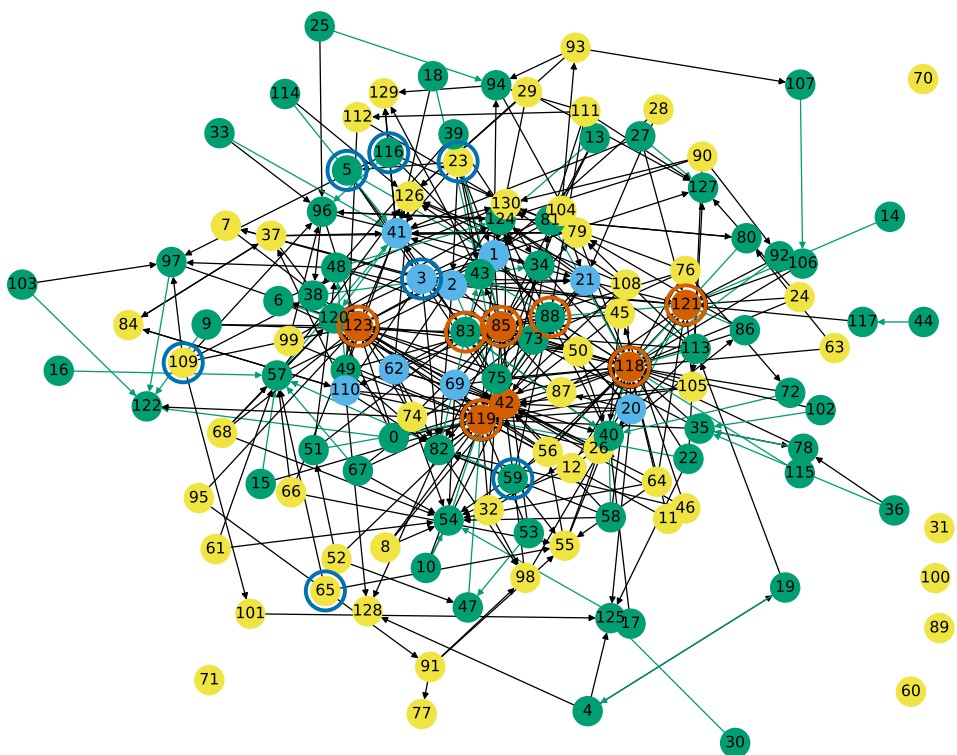

**Figure 4  Cross-shareholding network of Tehran Stock Market companies: Vermillion and sky blue nodes have respectively the highest in- and out-degrees.** Nodes circled in sky blue respectively vermillion have the highest entropic centrality under the current and reverse edge directionality.

121 (Adashare), 85 (NIKX1), 123 (Oilcopen), 42 (SA3A1), 119 (tamin org), and 118 (government). Nodes highlighted in sky blue have the highest out-degrees, which means that their shares are owned by many other companies (but possibly in small amounts). They are nodes 110 (BMEX1), 2 (CH12), 21 (FO041), 41 (GD021), 3 (GO02) with degree 7 and 69 (PFAX1), 1 (MADN), 20 (MS022) and 62 (PK061) with degree 8.

We next assign probabilities to edges: we use the edge weight, and fix the edge probability to be inversely proportional to its weight. Self-loops have a natural interpretation. Since the outgoing edges of node $j$ indicate the companies that are shareholders of $j$, the self-loop refers to $j$ still owning some of its own shares, and the amount is 100% minus what the other companies own (share ownerships with negligible amounts were not taken into account in the data set, so self-loops absorb these portions).

Table 1 lists the seven nodes with the highest entropic centrality. The interpretation of entropic centrality here is that we are looking at the nodes whose shares are "most diversely owned" in terms of their shares being owned by different companies, whose shares are in turn themselves owned by others. The economic fortunes of the company whose centrality is looked at thus also affects those of the other companies, and the entropic centrality thus indicates the impact a particular company's economic performance would have on the rest. We immediately see that this centrality measure is different from out-degree

**Table 1 Vertices with the highest entropic centrality, their scaled entropic centrality, where $f_u$ is the market size in percentage (their relative ranking is marked with subscript), their out-degrees (in the above part) or in-degrees (in the below part, for the graph with reverse edge directionality) and neighbors with the weight of the connecting edges.** Bold face indicates a high degree. Ranks with respect to alpha/Katz (AK) with $\alpha = 1$, PageRank (PR) weighted (W) and unweighted (U), and betweenness (B).

| no | ID | $C_{H,p}$ | $f_u$ | $f_u C_{H,p}$ | Out | Neighbors | A/K | PR(U/W) | B (W) |
|---|---|---|---|---|---|---|---|---|---|
| 23 | ARFZ1 | 3.1990 | 0.7153 | $2.2882_3$ | 6 | (**1**, 0.181), (**2**, 0.353), (5, 0.045), (43, 0.045) (**41**, 0.05), (85, 0.04) | 1/1 | 1/1 | 39 |
| **3** | GO02 | 3.0205 | 0.9635 | $2.9103_1$ | 7 | (**1**, 0.224),(**21**, 0.099) (82, 0.011),(7, 0.037) (42, 0.028), (43, 0.377) (121, 0.185) | 3/3 | 69/69 | 3 |
| 109 | IPAR1 | 2.9680 | 0.6874 | $2.0402_6$ | 5 | (96, 0.2),(101, 0.0963) (123, 0.173), (97, 0.078), (38, 0.139) | 22/22 | 7/4 | 39 |
| 59 | PRDS1 | 2.9541 | 0.7951 | $2.3488_4$ | 5 | (88, 0.011), (57, 0.668), (118, 0.054), (55, 0.05) (82, 0.011) | 14/14 | 18/26 | 39 |
| 65 | PK3A1 | 2.8857 | 0.6267 | $1.8084_7$ | 2 | (57, 0.42), (55, 0.206) | 71/71 | 30/12 | 39 |
| 5 | KNRX | 2.8817 | 0.9415 | $2.7131_2$ | 3 | (**1**, 0.8876),(**3**, 0.0209) (83, 0.033) | 26/23 | 67/37 | 31 |
| 116 | PRSX1 | 2.8273 | 0.8086 | $2.2861_5$ | 6 | (96, 0.648), (97, 0.049) (88, 0.071), (**41**, 0.011) (129, 0.01), (79, 0.017) | 8/7 | 12/9 | 39 |

| no | ID | $C_{H,p}$ | $f_u$ | in | Neighbors |
|---|---|---|---|---|---|
| **118** | gov | 5.3434 | 9.8 | 40 | (**85**,0.1737) |
| **119** | tamin org | 4.6989 | 5.7647 | 31 | (42, 0.0305) |
| **121** | Adashare | 4.4777 | 6.7827 | 17 | |
| 88 | BTEJ1 | 4.2916 | 0.7143 | 9 | (**85**, 0.474) |
| **123** | Oilcopen | 3.9154 | 3.5699 | 22 | |
| 83 | TMEL1 | 3.8099 | 0.4849 | 9 | |
| **85** | NIKX1 | 3.7693 | 0.9722 | 17 | |

centrality. We can however look at how they relate, by considering the role of out-degrees not only on the nodes but also on their neighbors. We observe that only node 3 has one of the highest out-degrees, however, nodes 23 and 116 still have high out-degrees, but also are connected to neighbors which have high out-degrees, in fact, node 23 which has the highest entropic centrality has three high out-degree neighbors. For nodes 109 and 59, they still have a relatively high out-degree. For 5, it has a fairly low out-degree, but out of the 3 neighbors, two have high degrees themselves. The case of node 65 is particularly interesting, since it has only two neighbors, namely 55 and 57. Neither of 55 nor 57 has a high centrality individually, but they together provide node 65 a conduit to a larger

subgraph than the individual transit nodes themselves do, illustrating the secondary effects of flow propagation.

The cross-shareholding network of Tehran Stock Market was analyzed in *Dastkhan & Gharneh (2016)*, where a closeness centrality ranking is shown to be almost identical to the degree based centrality one. The entropic centrality ranking in contrast manages to capture a different dynamics, by involving the spread of influence via flow propagation, together with a quantitative edge differentiation.

The above approaches ignore any other information such as the market size share of the organizations. Arguably, between two organizations with identical position in the graph, the one with larger market size may be deemed to have larger influence on the other nodes. This is modeled by the scaled entropic centrality $C_H(u)f_u$, where $f_u$ is the market size in percentage. We notice that this indeed creates a distinct relative ranking (indicated by subscript in Table 1, for example, ARFZ1 is ranked 3rd as per weighted entropic centrality). Particularly, among the top seven companies as per $C_H(u)$, we see that only PRDS1 continues to be in the same (fourth) rank. KNRX has the largest change in ranking, up from sixth to second. While the scaled entropic centrality ranking of the top two nodes are congruent with the ranking based solely on the scale factor (market size), we see that ARFZ1, which would be ranked 5th by market size, and first by solely network effect, is ranked 3rd when both factors are taken into account.

We compare the entropic centrality $C_{H,p}$ with respect to the alpha, Katz and PageRank centralities (using the reverse edge direction). Note that the unweighted graph has for largest eigenvalue $\lambda_1 \simeq 2.99715780$ (so $\frac{1}{\lambda_1} \simeq 0.3336494$). The ranking results for the most central entities from the Tehran stock exchange, and the overall Kendall tau rank correlation coefficients *Kendall (1938)* are reported in Tables 1 and 2 respectively. The Kendall tau coefficient indicates the rank correlation among a pair or ranked lists (see *Schoch, Valente & Brandes, 2017* for a discussion on why Kendall tau is preferred to Pearson). The entity 23 is outstandingly central with respect to all metrics but weighted betweenness. The alpha and Katz centralities yield very similar results, but they rank the entity 3 as third instead of second. They rank second the entity 20. A likely explanation could be that that 20 actually has a higher out-degree (and thus a higher in-degree in the reversed edge network) than entity 3. The top 7 most central entities have mostly a 0 betweenness (ranked 39), and are mostly ranked pretty low with respect to both versions of PageRank, weighted and unweighted. The most central entity for the weighted betweenness is 85, which is one of the most central with respect to in-degree (it has an in-degree of 18). Then 111 and 76 are second respectively for the unweighted and weighted PageRank. Entity 111 has out-neighbors 88,81,127,94,112 which become in-neighbors in the reversed edge graph, neither 111 itself nor its neighbors stand out by their degrees, however 76 has for in-neighbors in the reversed edge graph 72,73,130,85,118,76, and both 118 and 85 are very central with respect to in-degree, making it easier to explain why it is ranked high. Note that the assortativity coefficient of this graph is $\approx -0.01521584$, so this is a non-assortative graph, where high degree nodes do not particularly connect to neither high nor low degree nodes.

**Table 2  Kendall rank correlation coefficient $\tau_\kappa$ across the centralities for the Tehran stock exchange.**

|  | $C_{H,p}$ | Alpha ($\alpha = 0.1$) | PageRank (UW) $\alpha = 0.85$ | PageRank (W) $\alpha = 0.85$ | Katz ($\alpha = 0.1$) |
|---|---|---|---|---|---|
| $C_{H,p}$ | 0 | 0.2259 | 0.3185 | 0.3326 | 0.2252 |
| alpha | 0.2259 | 0 | 0.3171 | 0.3787 | 0.0047 |
| PageRank(UW) | 0.3185 | 0.3171 | 0 | 0.1317 | 0.316 |
| PageRank(W) | 0.3326 | 0.3787 | 0.1317 | 0 | 0.3778 |
| Katz | 0.2252 | 0.0047 | 0.316 | 0.3778 | 0 |

The Kendall rank correlation confirms that the entropic centrality differs from the other metrics not only to decide the most central vertices but also overall. The Kendall coefficient for unweighted betweenness has not been reported since only 38 vertices have a non-zero betweenness. This shows that the graph considered is far from being strongly connected. Overall, comparison points illustrate that the entropic centrality $C_{H,p}$ provides a new perspective not captured by the other algorithms.

We next consider the same graph but where edge directionality is reversed. A node becomes central if it owns diverse companies, which themselves may in turn own various companies. Since owning shares could be used to influence an organization's management, the entropic centrality based on reverse edges is thus a proxy indicator of how much control a specific entity has over the other entities in the market. Organizations with very high entropic centralities using either sense of edge direction could then be seen as probable candidates causing structural risks—be it by being 'too big to fail', or having too much control over significant portions of the market for it to be fairly competitive.

The nodes with highest entropic centrality are shown in Table 1. The most important one is the government: we expect it to be one of the most important players in Iran when it comes to owning shares in other companies (and yet not to appear in the list when the original edge direction is considered). We see a higher correlation between entropic centrality and in-degree than there was between entropic centrality and out-degree in Table 1. Among the seven most central nodes, five of them are having one of the highest in-degrees, the two most central nodes have themselves one high degree neighbor. Then node 88 has a fairly low in-degree, but it is connected to node 85.

In summary, the case study of the Tehran stock exchange network exhibits three important intertwined aspects of our model. Firstly, it is flexible. It seamlessly captures the effect of relationships, considered either in a binary fashion (just the *structure*), or quantified with relative strengths (the *skew* in strength of the relationships), while it can also accommodate information which is intrinsic to the node but somewhat disentangled from the network (used as a *scale* factor). Second, reversing the edge direction gives a dual perspective. Finally, we see that we obtain different results and corresponding insights, based on which variations of the model is applied for a specific study. Naturally, figuring it out the best variation is done on a case-by-case basis.

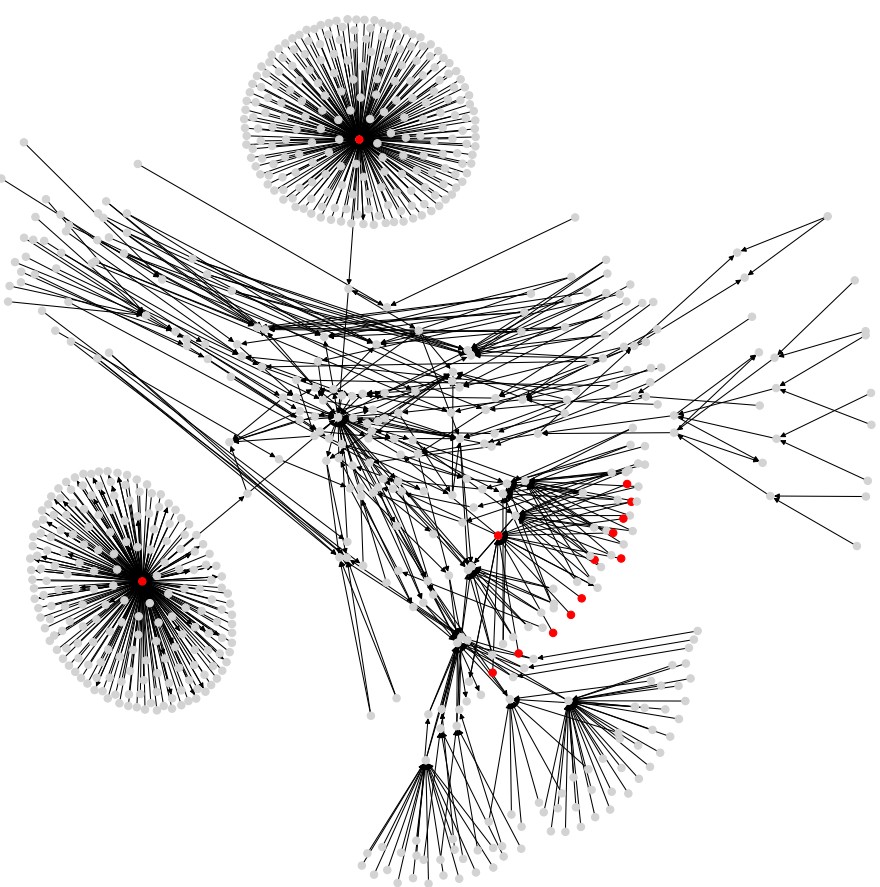

**Figure 5 A subgraph of the Bitcoin subgraph, which comprises only addresses that have non-zero entropic centrality.** Those in red are listed in Table 3, with the highest entropic centrality.

## A bitcoin subgraph

Our final case study is a subgraph of the Bitcoin wallet address network derived from Bitcoin transaction logs (see Fig. 5). Bitcoin is a cryptocurrency (*Nakamoto, 2008*), and transactions (buy and sell) among users of this currency are stored and publicly available in a distributed ledger called blockchain. User identities are unknown, but each user has one or many wallet addresses, that are identifiers in every transaction. Then one transaction record amalgamates the wallet addresses of possibly several payers and payment receivers, together with the transaction amounts.

To be more precise, consider two Bitcoin transactions $T_1$ and $T_2$. The transaction $T_1$ has $n$ inputs, from wallet addresses $A_1, \ldots, A_n$, of amounts $i_{11}, \ldots, i_{1n}$ respectively. The outputs, of amounts $o_{11}, \ldots, o_{1m}$ go to wallet addresses $C_1, \ldots, C_m$ respectively. The sum of inputs equals the sum of outputs and any transaction fees (say $\tau_1$), i.e., $|T_1| = \sum_{k=1}^{n} i_{1k} = \tau_1 + \sum_{l=1}^{m} o_{1l}$. For the sake of simplicity, we will ignore the transaction fees (i.e., consider $\tau_1 = 0$). A similar setting holds for transaction $T_2$, where the same wallet address $A_1$ appears again as part of the inputs, together with some wallet addresses $B_2, \ldots, B_{n'}$ which may or not

**Table 3** Addresses with highest entropic centrality in the Bitcoin subgraph above (with the respective relative ranks as per other centralities—alpha/Katz with $\alpha = 0.1$ (AK), PageRank (PR), weighted (W) and unweighted (U)) and with highest centrality when edges have reverse directionality below.

| Address | $C_{H,p}$ | $f_u$ | Out | $f_u C_H$ | AK | PR (U/W) |
|---|---|---|---|---|---|---|
| 3CD1QW6fjgTwKq3Pj97nty28WZAVkziNom | 8.6633 | 0.0473 | 2807 | 652 | 1 | 1/1 |
| 38PjB1ghFrD9UQs7HV5n15Wt1i3mZP8Wke | 5.7214 | 0.1961 | 382 | 481 | 3 | 14/14 |
| 3Eab4nDg6WJ5WR1uvWQirtMzWaA34RQk9s | 5.4339 | 0.1778 | 568 | 514 | 2 | 13/15 |
| 3MYqQJ5LbDe9U3drsaDprKxWobVZA3UgAw | 5.3316 | 0.9270 | 2 | 609 | 4 | 2/455 |
| 38mMQxz4knqfmecjLW3atdygfWxvvnJfg7 | 5.3316 | 0.9268 | 2 | 3 | 4 | 2/3 |
| 33XZf8Ys9sbqnAKynA4yBckyzwN3SEZaU7 | 5.3316 | 0.9254 | 2 | 9 | 4 | 2/10 |
| 3P4C7jpF1oxHgxqt4VgMRcCBEV3YEpaDUm | 5.3316 | 0.9224 | 2 | 7 | 4 | 2/8 |
| 3Fp5ejYY8FsJ6Y3kb377VRjJunTeUVYsuq | 5.3316 | 0.8966 | 2 | 2 | 4 | 2/3 |
| 3Q9SPyCN95szQUoQYgAHKgdhC3YnRsrFrW | 5.3316 | 0.8928 | 2 | 8 | 4 | 2/8 |
| 38A6nGSMR59WHVnj9gaJ2Cm62y9kFE318i | 5.3316 | 0.8908 | 2 | 5 | 4 | 2/6 |
| 3Ce7jUQn2RH5Ysdb4VvShoYymZLpkcqaAA | 5.3316 | 0.8877 | 2 | 10 | 4 | 2/11 |
| 364qbSJFhwkBgZnMuhmUHdczpaZNS2PmE6 | 5.3316 | 0.8832 | 2 | 1 | 4 | 2/2 |
| 3KDgKr3qov4Ws5WPnaA2RHjcE1N2UeVYs3 | 5.3316 | 0.8619 | 2 | 4 | 4 | 2/5 |
| 1NxaBCFQwejSZbQfWcYNwgqML5wWoE3rK4 | 5.3316 | 0.1175 | 2 | 6 | 4 | 2/7 |

| Address | $C_{H,p}$ | $f_u$ | in |
|---|---|---|---|
| 38PjB1ghFrD9UQs7HV5n15Wt1i3mZP8Wke | 7.6477 | 171.359 | 218 |
| 3Eab4nDg6WJ5WR1uvWQirtMzWaA34RQk9s | 7.5583 | 175.022 | 196 |
| 15hWpb3m5VXdn9KVsS4rSMnrQQJLhXVyN4 | 5.2649 | 7.504 | 17 |
| 1C7PDYzjRDqomyywDHEqx9huYoYQoGYgdV | 4.9876 | 3.949 | 31 |
| 1zksVRSDUuX2E5mMNvvbA9C4esfnvVdfA | 4.4176 | 0.494 | 2 |

intersect with $A_2,\ldots,A_n$. By design, Bitcoin transactions do not retain an association as to which specific inputs within a transaction are used to create specific outputs.

Suppose one would like to create a derived address network from some extract of the Bitcoin logs of transactions, that is a graph whose nodes are Bitcoin wallet addresses, and edges are directed and weighted. There should be an edge from address $u$ to address $v$ if there is at least one transaction where some amount of Bitcoin is going from $u$ to $v$. However as explained above, it is not always possible to disambiguate the input–output pairs. If the input amounts are particularly mutually distinct, and so are the output amounts, and there are input–output amounts that match closely, one might be able to make reasonable guess about matching a specific input to a specific output. In general, in absence of such particular information, one heuristic is to model the input–output association probabilistically. A common heuristic (*Kondor et al., 2014*) is to consider that based on transaction $T_1$ there is an edge from $A_1$ to each of $C_1,\ldots,C_m$. The same holds for transaction $T_2$. Thus in the derived address network, there will be an edge from $A_1$ to each of the $C_1,\ldots,C_m,D_1,\ldots,D_{m'}$. If some outputs $X,\ldots,Z$ are in common to both transaction outputs, there is a single edge between $A_1$ and each of the addresses $X,\ldots,Z$.

The derived address network gives us the graph to be analyzed, whose vertices are wallet addresses and edges are built as above. Originally, a given wallet address is sending

Bitcoins to possibly different output wallet addresses within one transaction, and the same wallet address may be involved in different transactions, with possible reoccurences of the same output addresses (this is the case of $A_1$ which is an input to both transactions $T_1$ and $T_2$ and $X, \ldots, Z$ appear as output transactions in both). In the split-and-transfer flow model, we can incorporate this information into the derived address network by assigning the probabilities $q(\{C_1, \ldots, C_m\}) = \frac{i_{11}}{i_{11}+i_{21}}$ and $q(\{D_1, \ldots, D_{m'}\}) = \frac{i_{21}}{i_{21}+i_{22}}$ with which the respective subsets $\{C_1, \ldots, C_m\}$ and $\{D_1, \ldots, D_{m'}\}$ of the set $\{C_1, \ldots, C_m, D_1, \ldots, D_{m'}\}$ of neighbors of $A_1$ are used. Other choices for $q(x)$ are possible, the rationale for this specific choice is to use a probability that is proportional to the amount of Bitcoin injected by $A_1$ in each of the transactions.

Edge weights in the derived address network are computed as follows. Let $|T_1| = \sum_{k=1}^{m} o_{1k}$ and $|T_2| = \sum_{k=1}^{m'} o_{2k}$ denote the total amounts involved in each of the transactions. For an edge between $A_1$ and $C_l$, which happens in $T_1$, it is $\omega_{C_1, \ldots, C_m}(A_1, C_l) = \frac{o_{1l}}{|T_1|}$, while for an edge between $A_1$ and $D_l$, which happens in $T_2$, it is $\omega_{D_1, \ldots, D_{m'}}(A_1, D_l) = \frac{o_{2l}}{|T_2|}$. We thus have

$$\sum_{l=1}^{m} \omega_{C_1, \ldots, C_m}(A_1, C_l) = \sum_{l=1}^{m} \frac{o_{1l}}{|T_1|} = 1, \sum_{l=1}^{m'} \omega_{D_1, \ldots, D_{m'}}(A_1, D_l) = \sum_{l=1}^{m'} \frac{o_{2l}}{|T_2|} = 1.$$

If some node pairs, and thus edges, repeat across transactions (for example, $A_1$ to $X, \ldots, Z$ in our example), these edge weights should cumulate in the overall derived address network. This is captured by the formula (2) which is here instantiated as

$$f_{A_1, X} = q(\{C_1, \ldots, C_m\})\omega_{C_1, \ldots, C_m}(A_1, X) + q(\{D_1, \ldots, D_{m'}\})\omega_{D_1, \ldots, D_{m'}}(A_1, X)$$
$$= \frac{i_{11}}{i_{11}+i_{21}} \frac{o_{1x}}{|T_1|} + \frac{i_{21}}{i_{11}+i_{21}} \frac{o_{2x}}{|T_2|}$$

where in transaction $T_1$ the output to address $X$ is $o_{1x}$, while it is $o_{2x}$ in transaction $T_2$.

In a departure from previous works that derive the address network in a manner explained above *Kondor et al. (2014)*, our graph model is able to retain the information that subsets of edges co-occur, or not, as displayed above. For that reason, the Bitcoin address network is a natural candidate (and in fact, part of the inspiration) for the general flow model with arbitrary splits and transfers as considered in this paper, where individual flows may go through a subset of outgoing edges.

For our experiments, we choose a Bitcoin subgraph appearing in the investigation of wallet addresses involved in extorting victims of Ashley-Madison data breach (see *Oggier, Phetsouvanh & Datta, 2018a* for accessing the data). It is obtained by extracting a subgraph of radius 4 (if the graph were undirected) around the wallet address 1G52wBtL51GwkUdyJNYvMpiXtqaGkTLrMv. While we would like to emphasize that we use here this Bitcoin graph to explore the entropic centrality model, it may still be worth mentioning that one identified suspect node from another of our study (*Phetsouvanh, Oggier & Datta, 2018*), namely node 15hWpb3m5VXdn9KVsS4rSMnrQQJLhXVyN4, has high enough entropic centrality to be listed (see Table 3 below) as a top entropic centrality node. Thus, the entropic centrality analysis can be used as a tool to identify nodes of interest, and to create a shortlist of nodes to be investigated further in detail, in the context of Bitcoin forensics.

**Table 4** Kendall rank correlation coefficient $\tau_\kappa$ across the centrality algorithms for the Bitcoin sub-graph dataset (excluding 3926 nodes which all had an entropic centrality score of zero).

| | $C_{H,p}$ | $f_u C_{H,p}$ | Alpha ($\alpha = 0.1$) | PageRank (UW) | PageRank (W) | Katz ($\alpha = 0.1$) |
|---|---|---|---|---|---|---|
| $C_{H,p}$ | 0 | 0.2331 | 0.2359 | 0.3421 | 0.3956 | 0.2342 |
| $f_u C_{H,p}$ | 0.2331 | 0 | 0.2135 | 0.3152 | 0.4337 | 0.2118 |
| alpha | 0.2359 | 0.2135 | 0 | 0.2652 | 0.4949 | 0.0032 |
| PageRank(UW) | 0.3421 | 0.3152 | 0.2652 | 0 | 0.4299 | 0.2652 |
| PageRank(W) | 0.3956 | 0.4337 | 0.4949 | 0.4299 | 0 | 0.4965 |
| Katz | 0.2342 | 0.2118 | 0.0032 | 0.2652 | 0.4965 | 0 |

Tables 3 and 4 compare the entropic centrality $C_{H,p}$ with other centralities. With respect to scaled entropic centrality, there is a large variation in the weightages associated with the edges, which has a significant impact on the relative rankings between scaled/unscaled entropic centralities. With respect to weighted betweenness, only three addresses are relevant, they are, with their respective in- and out-degree, 3Eab4nDg6WJ5WR1uvWQirtMzWaA34RQk9s (ranked 1, in-degree: 196, out-degree: 568), 38PjB1ghFrD9UQs7HV5n15Wt1i3mZP8Wke (ranked 2, in-degree: 218, out-degree: 382), and 3CD1QW6fjgTwKq3Pj97nty28WZAVkziNom (in-degree: 14, out-degree: 2807). The other addresses are ranked 69 (corresponding to a betweenness of 0). The graph has for largest eigenvalue $\lambda_1 \simeq 7.1644140$ and $\frac{1}{\lambda_1} \simeq 0.139578$. As with the previous cases, alpha and Katz centralities are very close to each other, they also agree more closely with $C_{H,p}$ on the most central addresses, but Table 4 shows that this is not the case in general. The trends shown by the Kendall rank correlation coefficient is similar to previous cases: there are more dissimilarities between PageRank and entropic centralities than between alpha/Katz and entropic centralities, but overall, entropic centralities give a different view point, as would be expected by extrapolating Borgatti's view point. The assortativity coefficient of the illustrated Bitcoin subgraph is $\approx -0.11914239$, suggesting a slight disassortativity. This is easily explained as an artefact of the way the subgraph was extracted (a small radius around a node), yielding a couple of hubs with nodes connected only to them (leaves). In this example, these leaves are having an entropic centrality influenced by having these hubs as their first neighbors.

As a last scenario, we consider the small network of Maine airports, with their connecting flights, for a total of 55 airports (see *Oggier, Phetsouvanh & Datta, 2018b* for accessing the data.). We created the network based on flights involving passenger for the period of January 2018 as per the data obtained from the United States Department of Transportation Bureau of Transportation Statistics website (https://www.transtats.bts.gov/DL_SelectFields.asp?Table_ID=292).

In Table 5, we synopsize the Kendall's tau coefficient $\tau_\kappa$. The lower the value of this coefficient, the closer (similar) two ranked lists are. We see that $C_{H,unif}$ produces results which are very similar to alpha and Katz centralities, but $C_{H,p}$ yields a reasonably distinct result instead. Furthermore, results from both PageRank applied to both weighted and unweighted graphs are most distinct both with respect to entropic centralities, as well

**Table 5** Kendall rank correlation coefficient $\tau_\kappa$ across the centrality algorithms for the airports network data.

|  | Uniform | Non-uniform | Alpha ($\alpha = 0.1$) | PageRank (UW) | PageRank (W) | Katz ($\alpha = 0.1$) |
|---|---|---|---|---|---|---|
| uniform entropic | 0 | 0.1730 | 0.0242 | 0.2087 | 0.2484 | 0.0242 |
| non-uniform entropic | 0.1730 | 0 | 0.1555 | 0.2228 | 0.2962 | 0.1555 |
| alpha | 0.0242 | 0.1555 | 0 | 0.2114 | 0.2390 | 0 |
| PageRank(UW) | 0.2087 | 0.2228 | 0.2114 | 0 | 0.2713 | 0.2114 |
| PageRank(W) | 0.2484 | 0.2962 | 0.2390 | 0.2713 | 0 | 0.2390 |
| Katz | 0.0242 | 0.1555 | 0 | 0.2114 | 0.2390 | 0 |

as the other existing centralities explored in our experiments. The assortative coefficient is $\approx -0.71478751$, this is thus a disassortative network. Indeed it contains two airports that serve as hubs, and small airports connected to it (or important airports whose edges have been cut when extracting the specific subgraph). In terms of entropic centrality, this corresponds to having small airports inheriting the influence of being connected to hubs.

## CONCLUSIONS

In this paper, we studied the concept of entropic centrality proposed by *Tutzauer (2007)*, which originally determined the importance of a vertex based on the extent of dissemination of an indivisible flow originating at it, by considering the uncertainty in determining its destination. We extended this concept to model divisible flows, which better reflect certain real world phenomenon, for instance, flows of money. In fact, one of the motivating scenarios that prompted us to study this model was to study the network induced by Bitcoin transactions—though, in the course of the work, and to validate the model, we also identified and analyzed other use cases, with arbitrary divisions of the flow. A previous work which considered only equitable divisions of the flow was shown to be a special case of the general model studied in this paper.

The flow based entropic centrality model bears in spirit some similarity with eigenvector based centrality measures in the sense that the importance of vertex node is determined by taking into account a transitive effect, namely, connections to a central vertex contributes to increase the centrality. We thus compared our approach with several representatives of this family, specifically alpha centrality, PageRank and Katz centrality. We observed that alpha and Katz centralities are closer to entropic centralities than PageRank (in terms of Kendall tau distance), but they are still fairly different. This could be extrapolated from the view point of *Borgatti (2005)*, which advocates to use path based centrality for transfer type of flow, and not eigenvector based centralities, which are best suited for duplication. This indicates that the new entropic centrality provides novelty not only in the principled manner in which it captures the phenomenon of divisible flows, but also in terms of the results and associated insights obtained from it.

### Funding
The work of Phetsouvanh Silivanxay was supported by a NTU Singapore scholarship for doing his PhD. The funders had no role in study design, data collection and analysis, decision to publish, or preparation of the manuscript.

### Grant Disclosures
The following grant information was disclosed by the authors:
NTU Singapore scholarship.

### Competing Interests
Anwitaman Datta is an Academic Editor for PeerJ.

### Author Contributions
- Frédérique Oggier conceived and designed the experiments, performed the experiments, analyzed the data, prepared figures and/or tables, performed the computation work, authored or reviewed drafts of the paper, approved the final draft.
- Silivanxay Phetsouvanh performed the experiments, prepared figures and/or tables, performed the computation work, approved the final draft.
- Anwitaman Datta conceived and designed the experiments, analyzed the data, prepared figures and/or tables, performed the computation work, authored or reviewed drafts of the paper, approved the final draft.

### Data Availability
The Tehran stock exchange dataset came from Dastkhan H, Gharneh NS. 2016. Determination of systematically important companies with cross-shareholding network analysis: a case study from an emerging market. International Journal of Financial Studies 4(3):13 DOI 10.3390/ijfs4030013 and the authors can be contacted at nshams@aut.ac.ir for the dataset.

The Bitcoin dataset from the Bitcoin transactions log is available at Oggier, Frederique Elise; Phetsouvanh, Silivanxay; Datta, Anwitaman, 2018, ''A 4571 node directed weighted Bitcoin address subgraph'', 10.21979/N9/IEPBXV, DR-NTU (Data), V1.

The Maine airport network dataset is available at Oggier, Frederique Elise; Phetsouvanh, Silivanxay; Datta, Anwitaman, 2018, ''Maine airport network in January 2018'', Available at 10.21979/N9/WM0K5W, DR-NTU (Data), V1.

### Supplemental Information
Supplemental information for this article can be found online at http://dx.doi.org/10.7717/peerj-cs.220#supplemental-information.

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
