# Peer review of "A split-and-transfer flow based entropic centrality"

_PeerJ Computer Science, doi:10.7717/peerj-cs.220_

## Round 0.1 · original submission · Major Revisions

Please take special attention to the concerns raised by Reviewer #1. Reviewer #1 recommended rejection, while both Reviewers #2 and #3 recommended Major Revisions, so clearly much work is needed.

Reviewer 1 ·

Basic reporting

In this work, authors propose a centrality metrics for complex networks based on the idea of entropy. Given one node, the centrality of which we want to calculate, paths starting from it are generated, and the entropy associated with the probability of reaching other nodes is evaluated. With respect to previous works, here a complication is introduced: paths (or flows) can split. In other words, suppose that a flow is exiting a node A, the latter being connected to B and C. Usually two possibilities are considered: either the flow passes through B, or it passes through C. In this work, on the other hand, the flow can be arbitrarily divided between B and C (such that, for instance, the 30% of it goes to B, and the 70% to C).

I have two major, and really important, negative comments about the proposed idea.

First of all, it is not clear why such complication is really needed, and whether the same problem could be solved by “renormalising” all link weights. Let me explain this with a slightly more complex example. Suppose the flow is exiting node A, which is connected to B and C. Also suppose that we have two possible “output patterns”: either the flow only goes to B with a 50% probability, or it goes to the pair (B, C) with a probability of 50%; also, in the latter case, the flow is split, such that half of it goes to B, and half of it to C. While this seems complex, it is easy to see that it is equivalent to say: 75% of the flow goes to B, and 25% to C. Interpreting this as probabilities, by starting from A we have a 0.75 probability of ending up in B, and 0.25 of ending up in C. Therefore, by renormalising the probabilities, we end up with a graph that can be analysed by more standard approaches - in a much more computationally efficient way. Thus, why do we need the framework proposed by the authors?

Secondly, and even provided that simpler solutions would not work, I fail to see when this approach would be really important. There are not many real-world situations that fulfil the proposed scenario; and actually, it is curious to see that this “split” mechanism is not used in the three examples provided by the authors! To illustrate, let us consider the case of the airport network. The full idea here presented, when using the example I reported in the previous point, would imply something like: “Airport A is connected to B and C. Passengers departing from A have two options: either take a flight whose destination is B, or take a second flight which will bring half of the passengers to B, and the other half to C.” Clearly this is quite weird!
I think that the authors completely fail at explaining why this tool, and its underlying hypotheses, is relevant in a real-world scenario.


Additionally, I have a set of smaller comments, which are reported below.

Table 1. The reported centrality is difficult to be analysed, as it is not compared with other standard approaches. For instance, I am pretty sure PWM would rank very high when considering a Betweenness or an alpha centrality. Authors should present this comparison, also to explain what is the difference between these metrics and the proposed one.

I find Fig. 1 quite useless, as the same example is then reported in Fig. 2 and following ones. Actually, the latter version is much better, as the figure reports more information (as probabilities, self-loops, etc.). Same for the text in the introduction; essentially, the same scenario is repeated several times. I don’t think that a very detailed description of this example is needed in the introduction, as it will be explained in Section 2.

The style of the text can be improved, and especially can be made more formal. I’ve lost count of the number of times the expression “say” is used in the introduction!

The bibliographic section could be made more complete. For instance, I find it surprising that no citations about complex networks theory are included. Note that the average reader of PeerJ may not be familiar with complex networks.

Line 188: “We notice the similarity with (1), which shows that the split-and-transfer entropic centrality is equivalent to the transfer entropic centrality, assuming the suitable computation of edge probabilities.” I find this sentence not really useful. According to definition 1, and if we consider just a “similarity”, every measure based on the Shannon entropy is also equivalent to the transfer entropic centrality!

Lines 209-210: bad formatting. A full stop is in the wrong position.

Experimental design

See previous section

Validity of the findings

See previous section

Additional comments

See previous section

Reviewer 2 ·

Basic reporting

The manuscript entitled "A split-and-transfer flow based entropic centrality" presents a new centrality measure which is based on a previously proposed entropic centrality.

The manuscript is well written and there are just a few typos. However there are some corrections to be performed in the text. Some of them are listed below.

- line 98: missing the number of the section which are being referenced
- line 164: "... corresponds to a set of edges (u,v) for v a neighbor"
- line 210: punctuation is missing before "The circled"

The main background and the introduction of the theme are well presented, however there are no references concerning related works in literature besides the one from which the proposed method is derived. Therefore, the introduction must be improved for what concerns other centrality measures and related works. What has been done so far?

The figures would be improved by using a vector format, like pdf or eps.

All the tables must be improved concerning their formatting. Captions must be located above the table. Table 1 should be divided into two tables.

In Example 1, the authors should assign a number to the sample figure.

Experimental design

The purpose of the study described in the manuscript is well defined and the main topic is relevant to the scope of the journal. The methodology is also well described and the several examples in the text illustrate very well the proposed approach.

The manuscript is well structured, presents a logical sequence of sections and a good summary of the obtained results. However, the conclusion part is missing. I also missed some discussion to what extent the proposed methods can be compared with other methods in literature.

Validity of the findings

It is missing a clear description of the data concerning its availability. The datasets can be freely accessed?

The supplementary material should contain a document explaining how to use the data and the available code.

Reviewer 3 ·

Basic reporting

The paper is designed to proposed a novel split-and-transfer flow for centrality entropy measure. Usually, a flow is considered to be indivisible instead to have an arbitrary split across choices of neighbors. Also, the authors show a scaled version of the proposal to allow probabilities on edges based on the application criteria.
Three case studies are presented to show the proposal.

The paper is very insteresting and show the proposal measure with very detailed images and accounts.

However, the work has several issues that should be modified prior to its publication.

1 - The writing mainly in the introduction section is sometimes very informal with many repetitions of words and to many uses of `say`. Try to substitute the word graph by network, or egde by connections just to improve the reading flow.

2 - The definitions of the terms are good mainly due to the draws in the paper.

3 - The number of sections at the end of the introduction are missing.

4- Wikipedia reference is not accurate for a scientific paper. Try to find another reference to enforce your sentence and the reference does not seems correct in the text.

Experimental design

The content of the paper is relevant and results prooves that the proposed model is applicable for real cases analysis.

Previous works focused on indivisible flow, while the study in question aims to compute centrality entropy in split-and-transfer flows.

Three case study are provided. the first in the main airport traffic, the second with shareholdings in tehran stock market and finally, the last case which analyses bitcoin transactions.

Results proves that the importance of nodes is the network proportional to the entropy centrality captured by the split-and-transfer flow from each node. Also, using a scaling function, the result is even better.

Since the definition of the flow and measures are explained with equation and examples, the problem seems to be reproductible.

Validity of the findings

The papers does contribute with the area and the results are very good.

A conclusion section could be added to summarize the proposal and reinforce its contribution with the results found the case studies.

Additional comments

- The work is very good in general, with good contributions to the science and very detailed explanations of the formulas.

- The network figures should be analysed to fit it better. The wikipedia reference is not very scientific and should be changed. Also, major revisions must be performed in the grammar (words repetitions and unformal english), latex references (sections numbers do not appears) to improve the reading.

Annotated reviews are not available for download in order to protect the identity of reviewers who chose to remain anonymous.

---

## Round 0.2 · Major Revisions

Even though some issues have been resolved, the main issue still remains. Reviewer #1 requires a better explanation on the usefulness of the proposed centrality measurement. I suggest the authors to fully address this important issue before submitting a new version.

Reviewer 1 ·

Basic reporting

While the paper has improved from the initial version, I still have problems with it, and specifically I fail to see the relevance of what proposed.

If I’m not wrong, the main assumption and novelty of this paper resides in the fact that flows are considered as divisible. From this we have the scenario of Fig. 3: in Fig. 3a one flow splits between v2 and v3, and in 3b another flow splits between v3 and v4.
Now, the authors state that: “We thus showed that the split-and-transfer entropic centrality is equivalent to a transfer entropic centrality, assuming the suitable computation of edge probabilities.” This is exactly my point: isn’t it always possible to reduce this complex split-and-transfer scenario to a more standard one, by changing the link probabilities?

Note that the authors previously answered:
“The high level observation made by the reviewer, that a renormalized link weight may capture the effects of arbitrary split of flows is correct. In fact, this is precisely what we demonstrate in this paper.”
But then, what is the point of the paper? Authors introduce a complex split-and-transfer centrality, but at the same time they demonstrate that this is equivalent to the standard transfer entropic centrality, under a link probability transformation. Then, what is the point of introducing a new metric?

Or, is there a scenario for which such transformation cannot be made? And thus where split-and-transfer centrality is the only viable solution? If is that so, authors should clearly describe it.

It seems to me that this paper is an overly complicated work to show something fairly simple…
But maybe I’m missing something… if so, please let me know!!

Experimental design

No comment

Validity of the findings

No comment

Additional comments

Please see my comments in the first section.

Reviewer 2 ·

Basic reporting

no comments

Experimental design

no comments

Validity of the findings

no comments

Additional comments

All the main issues raised by the reviewers were carefully addressed by the authors. The main purpose of the manuscript, the literature review and the related works are now clearer. In addition, the figures and tables are now well formatted. The manuscript was improved regarding the clarity of the presented methodology and results, as well as the text correctedness.

Reviewer 3 ·

Basic reporting

The paper proposes a new centrality measure that is designed to overcome the lack of metrics to compute centrality of not indivisible flows such as financial transactions.

In this new version, the authors added the comparative analysis with the results with other centralities measures.

Also, the introduction was improved with new background references (wikipedia reference was removed) and description of previous centralities with advantages and disadvantages according to the data analyzed.

Although one example was removed from the original manuscript, the authors have now related their work with other literature methods. Furthermore, the conclusion was improved.

Experimental design

The study is a novel measure for not indivisible flow, the authors took care to explain the proposal with detailed diagrams and showing the computation of the measure with examples.

However, Section 2.2 could be broken into paragraphs to help the readers to take time to understand each examples (Figures 3 a, b and c).

Validity of the findings

Two real cases were used to reinforce the application of the methodology. The data is explained in the supplementary data excluding the Tehran stock exchange dataset which is private.


The paper is suitable for publication in journal.

Additional comments

to be improved:

l.266 - The alpha and Katz centralities are giving, as expected, very similar results…

---

## Round 0.3 · Minor Revisions

Your review should address the following issues, as mentioned by one of the reviewers:

i) Authors should provide the computational cost to compute the entropic centrality.

ii) How the measure is influenced by the presence of degree-degree correlations (assortativity)?

iii) Include more datasets.

You may decide if the mentioned articles are adequate or not to be cited.
The final decision on your manuscript will not depend on whether you
cite or not those papers.

Reviewer 1 ·

Basic reporting

It seems we are at a stalemate here, as my main concern (and it has not changed) is about the relevance of the new metric - I still don't see why it is so important to use it. Nevertheless, I went back to the requirements of PeerJ, and referees are NOT asked to evaluate "potential impact; degree of advance; possible readership etc.". Therefore, I will not block further the publication of this manuscript.

Experimental design

No additional comment

Validity of the findings

No additional comment

Additional comments

No additional comment

Reviewer 4 ·

Basic reporting

The introduction has very long paragraphs. This is not common in scientific writing.

The paper has only 6 references from the last five years, authors could include more recent references.

Nowadays exist several centrality measures, authors cite only six measures: degree, closeness, betweenness, eigenvector, Katz and PageRank,
More measures could be cited and related to the proposed work since the focus is a new centrality. For example, see this recent work:

- Influencers identification in complex networks through reaction-diffusion dynamics
Flavio Iannelli, Manuel S. Mariani, and Igor M. Sokolov

Experimental design

Authors should provide the computational cost to compute the entropic centrality.

How the measure is influenced by the presence of degree-degree correlations (assortativity)?

How the measure is related to the diffusion process, like epidemics? (see reference above)

Employ more graph-datasets and provide statistical analysis of the results.

Validity of the findings

Some authors show there exists a high correlation among centrality measures, for example:
- Correlations among centrality indices and a class of uniquely ranked graphs
David Schocha, Thomas W. Valente, Ulrik Brandes

- A multi-centrality index for graph-based keyword extraction
Didier A.Vega-Oliveros, Pedro Spoljaric Gomes, Evangelos E. Milios, Lilian Berton

To analyze the correlation authors should compute also the Pearson coefficient and employ:
1) more measures especially using random walk
2) more datasets, two datasets are not enough
3) compute the correlation distribution among the datasets and measures.

---

## Round 0.4 · Minor Revisions

Please respond to the final minor comments below

Reviewer 4 ·

Basic reporting

no comment

Experimental design

no comment

Validity of the findings

no comment

Additional comments

Authors included two recent references, one more dataset, calculated the assortativity and mentioned the time complexity as suggested.
It is not clear if the time complexity is the same as DFS, authors could mention O(V+E) in this case.

Some suggestions to improve readability:
1) URL could be included as a footnote.
2) the addresses in Bitcoin network are very long char, authors could use a simple code to mention each one in the text.
3) Tables could be standardized (about the information and font size).
4) standardize the number of decimal digits, in some tables, it is four, five or six.
5) if the authors have lack of space they could remove fig. 5, it does not aggregate much information in the work.

---

## Round 0.5 · accepted · Accept

Your article is now Accepted.